# A novel serum miRNA-pair classifier for diagnosis of sarcoma

Zheng Jin[1], Shanshan Liu[1], Pei Zhu[1], Mengyan Tang[1], Yuanxin Wang[1], Yuan Tian[2], Dong Li[1], Xun Zhu[1], Dongmei Yan[1]*, Zhenhua Zhu[3]*

1 Department of Immunology, College of Basic Medical Sciences, Jilin University, Changchun, Jilin Province, China, 2 Key Laboratory for Molecular Enzymology and Engineering, Jilin University, Changchun, Jilin Province, China, 3 Department of Orthopaedic Trauma, The First Hospital of Jilin University, Changchun, Jilin Province, China

* dmyan@jlu.edu.cn (DY); zhuzhenhua93@hotmail.com (ZZ)

**Data Availability Statement:** Data are publicly available from the Gene Expression Omnibus (GEO) at www.ncbi.nlm.nih.gov/geo/ (accession code GSE124158) and from the ArrayExpress repository at www.ebi.ac.uk/arrayexpress/

## Abstract

Soft tissue sarcomas (STS) is a set of rare malignant tumor originated from mesoderm. For the prognosis of sarcoma, early diagnosis is important, however, currently no mature and non-invasive method for diagnosis exists. MicroRNAs (miRNAs) are a class of noncoding RNAs and their expression varies greatly, especially during tumor activity. The purpose of this study was to construct a predictive model for the diagnosis of sarcomas based on the relative expression level of miRNA in serum. miRNA array expression data of 677 samples including 402 malignant sarcoma samples and 275 healthy samples was used to construct the prediction model. Based on 6 gene pairs, random generalized linear model (RGLM) was constructed, with an accuracy of 100% in the internal test dataset and of 74.3% in the merged external dataset in prediction whether a serum sample was obtained from a sarcoma patient, with a specificity of 100% in the internal test dataset and 90.5% in the external dataset. In conclusion, our serum miRNA-pair classifier has the potential to be used for the screening of sarcoma with high accuracy and specificity.

## Introduction

In general, sarcomas are divided into bone and soft tissue sarcomas, both of which have many subtypes [1]. Advances in adjuvant chemotherapy and surgical techniques have provided additional options for the treatment of sarcomas. However, high-grade sarcomas are prone to recurrence and metastasis and once metastasized, the death rate can increase to 50% [2]. CT and MRI imaging and biopsy are routine diagnostic methods for sarcomas, but the high costs and invasive nature of the approach are not conducive to the screening of sarcomas. Development of simple and convenient screening method is of critical importance for the treatment and prognosis of sarcomas.

MicroRNAs (miRNA) are short ($\sim$ 22 nucleotides) non-coding RNA molecules that regulate gene expression at the post-transcriptional level, and have critical functions across various biological processes [3]. Furthermore, miRNAs showed higher accuracy than messenger RNA in classifying poorly differentiated tumors in a study of 334 samples [4]. Certain subsets of

(accession codes E-MTAB-3273, E-MTAB-3888, E-MTAB-5126).

**Funding:** The work was supported by the grants from the National Natural Science Foundation of China (No. 81571530 and 81871245).

**Competing interests:** The authors have declared that no competing interests exist.

miRNAs are secreted from cancer cells into the extracellular space via multiple mechanisms, such as microvesicle-mediated pathways [5, 6]. In light of these biological features of miRNAs, extracellular miRNAs also known as circulating miRNAs showed potential to be used as diagnostic markers for a variety of tumors [7]. Indeed, in a recent study by Asano *et al* [8], it was shown that the expression level of 7 miRNAs exhibited good performance to determine whether serum samples were from sarcoma patients or healthy donors as indicated by an area under the ROC (receiver operating characteristic) curve (AUC) value of 98%. Although these findings indicate significant progress, due to differences in miRNA detection methods, the accuracy of prediction models that are based on miRNA expression levels may fail. Gene pairs based on relative expression level, eliminating a batch effect may be an alternative choice [9].

In this study, we constructed gene pairs according to the relative expression levels of miRNAs, and identified a novel miRNA pairs-based classifier in the screening of sarcoma.

## Materials and methods

### Data source

Data processing steps are shown in the flow chart (Fig 1). Gene Expression Omnibus (GEO) and ArrayExpress, the two largest sequencing data platforms were searched for data retrieval. Datasets should fulfill the following criterial to be adopted: 1. must include miRNA array or sequencing data of serum samples; 2. serum must be taken from sarcoma patients or healthy volunteers.

### Screening miRNAs and constructing gene pairs

For screening miRNAs and constructing gene pairs, we first screened miRNAs and selected miRNAs with sufficient expression in sarcoma. In this study, missing expression values were filled using K-nearest neighbors (KNN) algorithm. Then, miRNAs with an expression higher than 8 (log2 scale) in half of the samples of GSE124158 were selected. After that, samples from GSE124158 including malignant sarcoma patients and healthy subjects were randomly allocated to training group and test group at a ratio of 3:1. In the training dataset, t-test was used to test the statistical significance of each miRNA between healthy and sarcoma samples. miRNAs with p value less than 0.05 and effect value ranked in the top 250 were selected as candidate miRNAs. The expression level of candidate miRNAs underwent pairwise comparison to generate a score for each gene pair. If the first gene (G1) of a gene pair was smaller than the second in a single sample, then the value of this gene pair in this sample was set to 1, in other cases, it was set to 0. According to the above rules, we constructed the gene pairs—samples matrix. To ensure the prediction efficiency of the model, gene pairs which had a 1(or 0) in most samples (>90%) of training dataset were removed. Then, in the training dataset t-test was used to test the statistical significance of each gene pair between healthy and sarcoma samples in the training dataset. miRNAs with p value less than 0.05 and effect value ranked in the top 80 were selected as candidate gene pairs for prediction model construction.

### Classifier construction and validation

Random generalized linear model (RGLM) is a highly accurate and interpretable ensemble predictor that shares the advantages of a random forest (excellent predictive accuracy, feature importance measures, out-of-bag estimates of accuracy) with those of a forward selected generalized linear model (interpretability) [10]. Variables were randomly selected into 100 bags, and in each bag, variables were filtered by correlation test and stepwise method. After that, generalized linear model (GLM) is constructed in each bag by using the filtered variables.

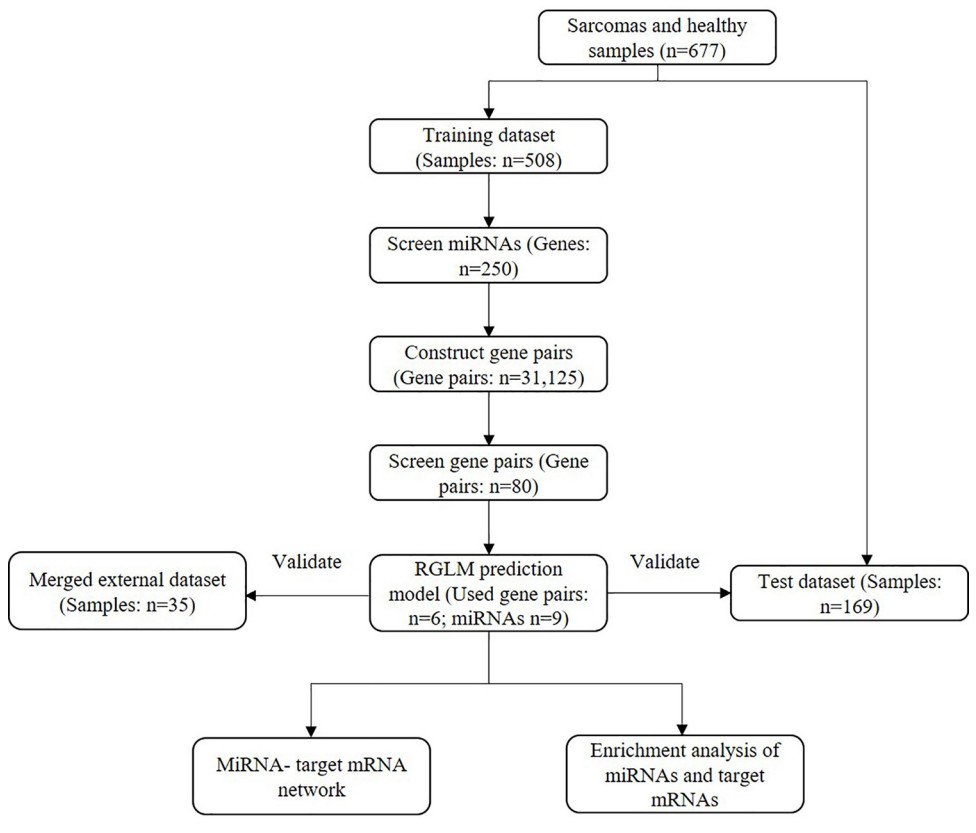

**Fig 1. Flow diagram of this study.**

When test dataset is predicted, the model will use the voting method to synthesize the prediction results of 100 independent GLM models to give the final prediction results. Using the screened gene pairs, RGLM was used to construct a prediction model that determined whether samples were healthy samples or sarcoma samples. Then, using "thinRGLM" function, gene pairs that occurred the most in the 100 GLMs were reserved and the thinned RGLM model was constructed based on that. Meanwhile, the reduction in prediction accuracy was negligible compared to the original RGLM model. Subsequently, the classifier was tested in the internal test dataset and external dataset. Currently, no biological diagnostic indexes for soft tissue sarcomas are commercially available. To test the predictive efficiency of this model, the model was compared with the original prediction model of Asano [8].

## miRNA network and function enrichment analysis

Using the miRNet platform [11] (https://www.mirnet.ca/), miRNA targets were predicted and a correlated network was constructed. Using the online databse STRING (https://string-db.org/) [12], gene ontology (GO) function enrichment analysis was performed based on the predicted target mRNAs.

## Statistical analysis

All the statistical analyses were performed using R version 3.6.3 (R Foundation for Statistical Computing, http://www.R-project.org) and associated packages. The caret package (v 6.0) was used to divide the samples into training/test partitions in GSE124158 at a ratio of 3:1 according

to the type of samples. The DMwR package (v 0.4.1) was used to fill in NA values with the values of the nearest neighbors. T-test between groups was conducted using package genefilter (v 1.68.0). In all statistical analysis, p<0.05 was considered statistically significant. In this study, accuracy, specificity and sensitivity were used to evaluate the predictive effect of the model. $accuracy = \frac{TP+TN}{TP+TN+FP+FN}$ , $specificity = \frac{TN}{FP+TN}$ , $sensitivity = \frac{TP}{TP+FP}$ (TP: True positive; TN: True negative; FP: False positive; FN: False negative).

## Results

### Characteristics of datasets

A total of 677 serum samples, including 402 malignant sarcoma samples and 275 healthy controls and associated basic clinical information was downloaded from GEO, serial number GSE124158 [8]. In addition, external test data sets E-MTAB-3273 [13], E-MTAB-3888 and E-MTAB-5126, containing 10 synovial serum samples, 6 liposarcoma serum samples, 5 leiomyosarcoma serum samples, and 14 healthy controls were downloaded from the Array Express. Detailed clinical information of GSE124158 and merged external dataset are presented in Table 1 and S1 Table.

### Gene screening and the construction of gene pairs

At the threshold of 8 (log2 scale), 362 miRNAs were identified as abundant miRNAs. Using t-test, 250 miRNAs were screened out for construction of the gene pairs. The genes were paired to produce a total of 31,125 gene pairs. Next, gene pairs were filtered to remove gene pairs with consistent values of 0 or 1 in most of the samples (90%). Then using t-test, a total of 80 gene pairs were selected to construct the prediction model.

### Prediction mode

In the RGLM prediction model, the gene pairs are lessened on the premise of predicting accuracy. Finally, a total of 6 gene pairs, containing 9 miRNAs were used in the final prediction model. Gene pairs included hsa-miR-378c, hsa-miR-383-3p, hsa-miR-454-5p, hsa-miR-4740-5p, hsa-miR-5007-3p, hsa-miR-380-5p, hsa-miR-499b-3p, hsa-miR-571 and hsa-miR-518a-3p, which are listed in Table 2. RGLM model showed 100% accuracy, specificity and sensitivity in predicting whether samples were healthy samples or sarcoma samples in the internal test dataset. The gene pair-based prediction model showed an accuracy of 74.3% in the outside test

**Table 1. Clinical information of dataset GSE4158.**

| Characteristics | Details | |
|---|---|---|
| | Sarcoma | Health |
| Age (median±sd) (years) | 48±22 | 51±12 |
| Gender (N) | | |
| Male | 244(60.7%) | 150(54.5) |
| Female | 158(39.3%) | 125(45.6%) |
| Stage (N) | | |
| Stage I | 36 (9.0%) | |
| Stage II | 125(31.1%) | |
| Stage III | 137(34.1%) | |
| Stage IV | 100(24.9%) | |
| Unknown | 4(1.0%) | |

**Table 2. Gene pairs for RGLM prediction model.**

| Gene pairs | Gene 1 | Gene 2 |
|---|---|---|
| 1 | hsa-miR-378c | hsa-miR-380-5p |
| 2 | hsa-miR-378c | hsa-miR-499b-3p |
| 3 | hsa-miR-383-3p | hsa-miR-571 |
| 4 | hsa-miR-454-5p | hsa-miR-571 |
| 5 | hsa-miR-4740-5p | hsa-miR-5007-3p |
| 6 | hsa-miR-5007-3p | hsa-miR-518a-3p |

dataset with a specificity of 90.5% and a sensitivity of 50.0% (Table 3). In our prediction model, 9 samples were predicted as normal samples and 26 samples were predicted as STS samples. While, all the 35 samples were predicted to be healthy in Asna's model in the external dataset.

## miRNA network and function enrichment

Five of the 9 miRNAs have experiment validated target information in the miRNet platform. Based on the miRNA-mRNA target information of miRNet, a miRNA network containing 247 nodes and 248 edges was constructed (Fig 2, S2 and S3 Tables). Four miRNAs (hsa-mir-378c, hsa-mir-454-5p, hsa-mir-571 and hsa-mir-499b-3p) linked with their respective targets and shared common targets. Function enrichment analysis revealed that targets mRNAs mainly play a role in metabolic process such as protein metabolic process, cellular macromolecule metabolic process, etc. (Fig 3). Furthermore, MAPK and FoxO signaling pathways were also significantly enriched (S4 Table).

## Discussion

Traditional methods of detecting tumors are usually harmful to the human body. It is well known that miRNAs are involved during development and physiological processes, and their disorders may lead to the development of several diseases [14]. Because miRNAs can reflect pathological processes, they have been considered useful biomarkers for diagnosis and pathogenesis, as well as for classification of different cancer types [15].

With the development of high-throughput sequencing technology, an increase in genetic testing approaching has been applied in clinical medicine to diagnose or assess prognosis. For instance, Li *et al* developed an individualized immune signature, which can estimate the prognosis in patients with non-small cell lung cancer in an early stage [16]. In a previous study, a gene pairs-based prognostic signature has been used for estimating the prognosis of gastric cancer [17], and relapse-free survival of colorectal cancer [18]. In our study, we aimed to investigate the diagnostic ability of gene pairs for sarcomas.

**Table 3. Predictive accuracy of the gene pair classifier.**

| Datasets | Accuracy | Specificity | Sensitivity | PPV | NPV |
|---|---|---|---|---|---|
| Train dataset (N = 508) | 99.8% | 99.7% | 100% | 99.5% | 100% |
| Internal test dataset (N = 169) | 100% | 100% | 100% | 100% | 100% |
| External test dataset (N = 35) | 74.3% | 90.5% | 50.0% | 77.8% | 73.1% |

*PPV* positive predictive value, *NPV* negative predictive value

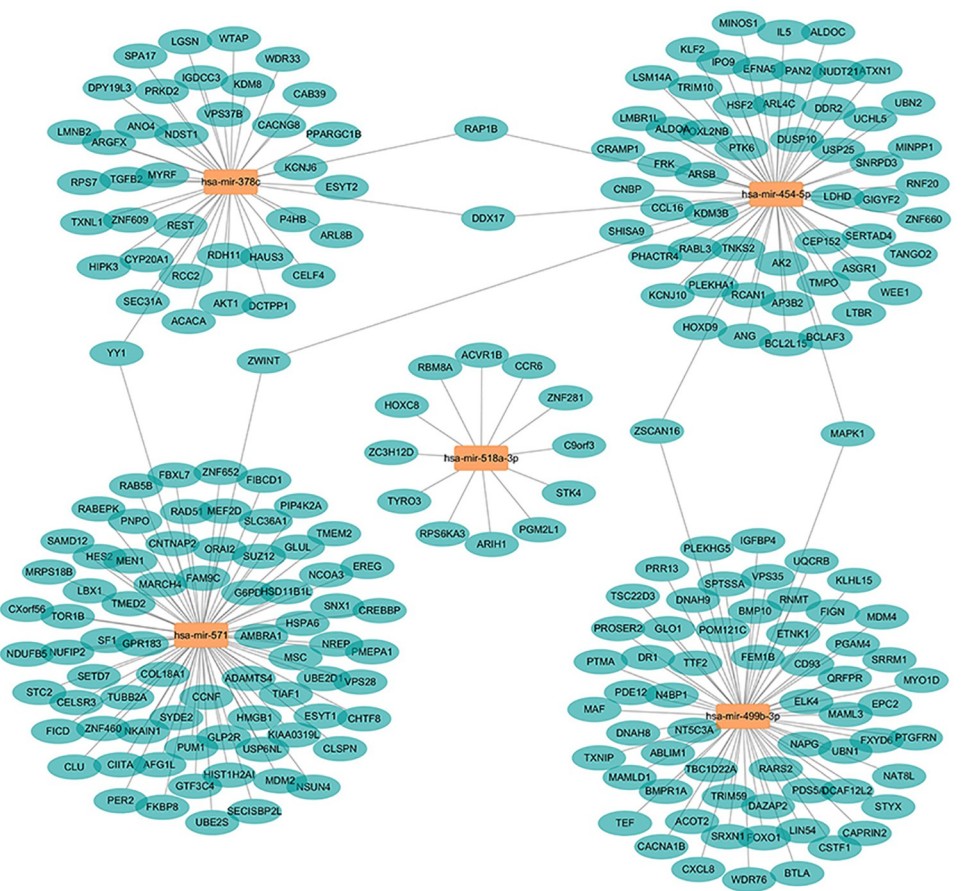

**Fig 2. miRNA network based on the miRNAs in the prediction model.**

Due to technical and algorithmic differences, models based on gene expression levels of one dataset may not be accurate to predict another dataset, which is especially true for datasets from different sequencing platforms [9]. Asano N *et al* developed an index based on gene expression levels that could distinguish sarcomas from benign or healthy samples [8]. In our models, the accuracy of the prediction was up to 100% in the internal test dataset. In addition, we applied our model to external datasets and achieved an accuracy of 74.3% with a specificity of 90.5%. In contrast, the prediction model "Index VI" which based on the expression level of 7 miRNAs failed to classify the samples. This may be caused by large differences in gene expression signals between different chip platforms. But using the idea of gene pairs can reduce this effect because there is no need to consider the specific expression level of genes.

As a preliminary exploration of the function of miRNAs in the prediction model, we constructed a miRNA-mRNA network using miRNA and their target information. Enrichment analysis revealed that the network may be related to metabolic processes, but further confirmation is needed.

Blood samples are easily obtained in physical examination, and further analysis of blood samples to identify tumor patients is of positive significance for the treatment and prognosis of tumor. Based on this, we built a classifier based on gene pairs, which showed good accuracy and specificity. Due to limited data, the reliability of this classifier needs to be tested in more samples.

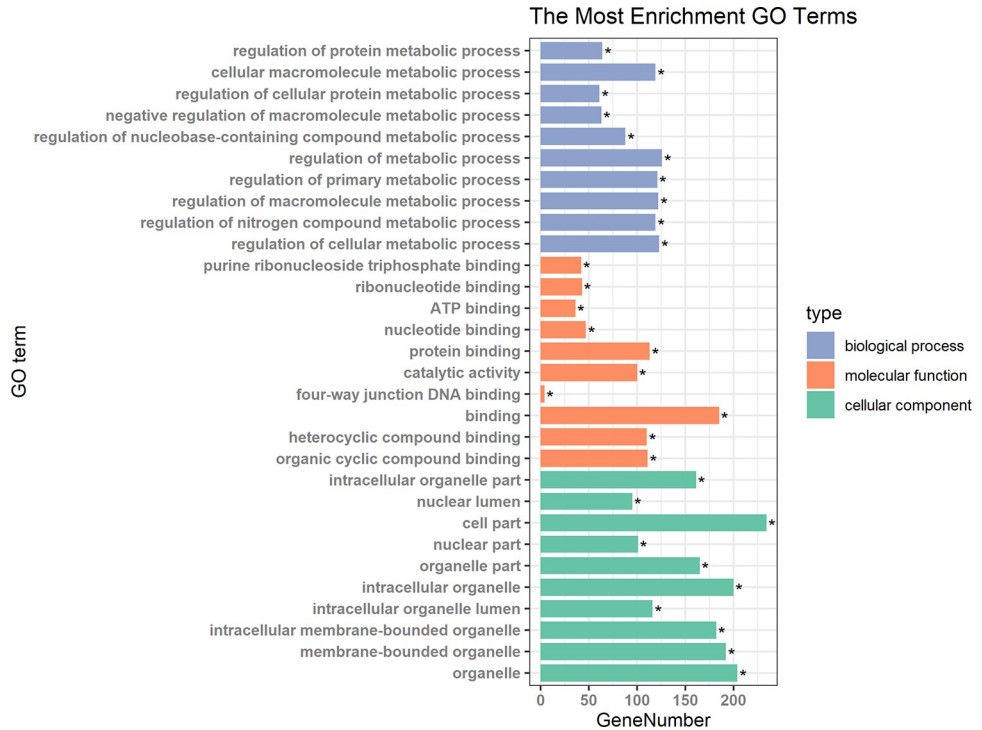

**Fig 3. Biological process enrichment results of the miRNA network.**

## Conclusions

In our study, we identified a novel gene pairs based classifier that is promising in the screening of sarcomas.

## Supporting information

**S1 Table. Clinical characteristics of the external datasets.**
(DOCX)

**S2 Table. Nodes information of miRNA-mRNA network.**
(DOCX)

**S3 Table. Edges information of miRNA-mRNA network.**
(CSV)

**S4 Table. GO and KEGG enrichment items of miRNA network.**
(XLSX)

**S1 Codes. Codes and original data for reproducing the results in this study.**
(ZIP)

## Author Contributions

**Data curation:** Zheng Jin, Shanshan Liu.

**Formal analysis:** Shanshan Liu.

**Funding acquisition:** Dongmei Yan.

**Investigation:** Pei Zhu.

**Methodology:** Zheng Jin, Mengyan Tang, Dong Li.

**Validation:** Mengyan Tang, Yuanxin Wang, Yuan Tian, Dong Li, Xun Zhu.

**Visualization:** Zhenhua Zhu.

**Writing – original draft:** Zheng Jin.

**Writing – review & editing:** Dongmei Yan, Zhenhua Zhu.

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
