## [Decision Letter · Decision Letter 0]

5 Mar 2020

PONE-D-19-34131

Development and validation of individualized miRNA signatures for sarcoma

PLOS ONE

Dear Dr. Zhu,

Thank you for submitting your manuscript to PLOS ONE. After careful consideration, we feel that it has merit but does not fully meet PLOS ONE’s publication criteria as it currently stands. Therefore, we invite you to submit a revised version of the manuscript that addresses the points raised during the review process.

We would appreciate receiving your revised manuscript by Apr 19 2020 11:59PM. To enhance the reproducibility of your results, we recommend that if applicable you deposit your laboratory protocols in protocols.io, where a protocol can be assigned its own identifier (DOI) such that it can be cited independently in the future. For instructions see: http://journals.plos.org/plosone/s/submission-guidelines#loc-laboratory-protocols

We look forward to receiving your revised manuscript.

Kind regards,

David M Loeb

Academic Editor

PLOS ONE

Journal Requirements:

3. Please include a copy of Table 1 and 2 which you refer to in your text on page 7.

Reviewers' comments:

Reviewer's Responses to Questions

**Comments to the Author**

1. Is the manuscript technically sound, and do the data support the conclusions?

Reviewer #1: Partly

Reviewer #2: Partly

Reviewer #3: Partly

2. Has the statistical analysis been performed appropriately and rigorously? 

Reviewer #1: I Don't Know

Reviewer #2: No

Reviewer #3: I Don't Know

3. Have the authors made all data underlying the findings in their manuscript fully available?

Reviewer #1: Yes

Reviewer #2: Yes

Reviewer #3: Yes

4. Is the manuscript presented in an intelligible fashion and written in standard English?

Reviewer #1: No

Reviewer #2: Yes

Reviewer #3: Yes

5. Review Comments to the Author

Reviewer #1: In this paper by Jin et al, the authors evaluated circulating microRNA signatures from patients with soft tissue sarcomas (STS) in order to develop predictive models of diagnosis. The investigators evaluated more than 880 sarcoma samples and determined 13 gene pairs that could be used for diagnosis with an AUC of 100% in their initial set, but only 68% in the external validation set. Of note, they also determined that the miRNA gene pairs showed enrichment for functional roles in cell cycle regulation. They concluded that their serum miRNA pairs can be used for the diagnosis of STS with high degree of accuracy and sensitivity. Overall, these authors have developed a non-invasive and potentially clinically applicable prediction model using serum microRNA signatures. However, there are some concerns regarding their approaches and conclusions, which are mentioned below.

Lines 59-60: Please further explain exactly what is meant by the statement: “detecting the level of miRNAs in body fluids, the state of somatic tumor cells can be inferred..”

Lines 106-107: What does it mean for the miRNAs to be “paired in the order from front to back”?

It is still not clear how the miRNAs were paired. It would be helpful to more of the general audience to further explain this process.

Lines 159-160: How were the 96 gene pairs, from more than 11,0000 pairs, selected to construct the prediction model.

But then in lines 163-164 it is stated that a total of 13 gene pairs, containing 16 miRNAs were used. All of this process is very confusing and difficult to follow for the reader.

Lines 173-174: It would be informative to provide more information regarding the other models used for prediction development and the comparison with this particular model.

Finally, all of the figures are extremely difficult to review. The clarity is extremely poor.

Reviewer #2: In this manuscript, the authors describe a miRNA gene pairs-based index to differentiate patients who have bony lesions from healthy controls. The concept is interesting and such biomarker development has tremendous promise in the sarcoma field. In particular, it is intriguing that the predictive signature was consistent across all stages of disease. However, there are critical weaknesses that need to be addressed for this manuscript to be considered for publication.

Major critiques:

1. The patient cohort is comprised of tumors listed as benign, intermediate, and malignant, all of which (individually and collectively) can be distinguished from healthy controls. Benign bone lesions (40+% in the cohort), by definition, are not sarcomas, and therefore the major conclusion that this model can distinguish sarcoma is not substantiated by the data. The signature can impressively distinguish all bone lesions from healthy controls in the test/training set, and there is only a minimal ability (figure 3C) to distinguish malignant vs benign. The authors may want to contend that a benign bone lesion may undergo malignant transformation, and therefore distinguishing benign vs intermediate vs malignant is not critical in their model.

2. The title, abstract, and body of the manuscript state that the approach has been validated, yet the validation in a very small (25 patient) external dataset did not perform nearly as well as the test/training set (65%). This is not unusual performance for such assays when there are differences in sample processing, chips, etc, though it fails to achieve the bar of validation as highlighted in the title and elsewhere. Of interest, the external dataset contained patients with varied sarcomas (i.e., malignant tumors), yet the test/training sets were composed of patients who had malignant and benign lesions. Separating out benign from malignant, and possibly eliminating patients with “intermediate” tumors from your analysis may lead to a more robust predictive model.

3. The broad title focused on miRNA signatures for sarcoma does not reflect the findings in the manuscript since it’s not only sarcoma that was differentiated from healthy controls, but rather bone lesions that may or may not be sarcoma.

Minor critique:

1. In the introductory background about the promise of early detection of patients with sarcoma from healthy controls, it would be more compelling to describe practical application, such as a patient presenting with a bone lesion and this model signature can differentiate benign from malignant. Further clinical application of such an assay could be as a post-treatment screening test for patients with sarcoma to identify early recurrence of disease. However, based on the major critiques above, this may not be a potential application of the described research.

Reviewer #3: This manuscript utilizes an in silico analysis of miRNA database to develop a predictive model for the detection of sarcomas from serum samples.

Some suggestions for the authors consideration:

1. Line 160, the authors states that 96 gene pairs were selected to construct the prediction model, however in the prediction model the authors state that a total of 13 gene pairs were used. The authors should consider in explaining further as to how they further narrowed down the gene pairs from 96 to 13 and perhaps also why they felt that was necessary.

2. In Line 178, the authors state that the gene pair prediction model showed an AUC of 68.4% in the outside data set. This is a large drop in the ability of the model to predict sarcoma when using an external data set. In some literature an AUC of 0.7-0.8 is considered acceptable and 0.8-0.9 is considered excellent. The authors conclusion that their model has high accuracy and sensitivity does not seem to be consistent when they tested their model to the external data set. The AUC of 100 in the internal data set, may simply denote that differences in the methodologies used to measure the serum miRNA may have a larger impact than the gene-pair model they developed. The authors should explain the implications of the drop in the AUC in regards to the broader applicability of the model.

3. The authors conclude that their test is very sensitive. It would be very important for the authors to better state the stage of the disease when the serum samples were obtained. If the serum samples are all obtained when the patients have active detectable disease, what would be the applicability of this model? Knowing that a patient had a "sarcoma" may not necessarily be practically beneficial, as a biopsy would still be required to define histological type. In lines 239-240, the authors state that this model "is promising in the diagnosis of sarcoma". However, that information, even if the AUC was better than 68% would not be sufficient for making treatment decisions. How well does this model work in the minimal residual disease state? This may have broader applicability to replace serial imaging? The authors may consider narrowing their conclusions to state that the model they have generated was outstanding in their internal test data set, but had significant reduction in the AUC when exposed to an internal data set in detecting sarcomas in patients with known active disease.

4. in the methods section for the sake of completeness, the authors may consider adding on which platform the statistical analysis was performed.

6. PLOS authors have the option to publish the peer review history of their article (what does this mean?). If published, this will include your full peer review and any attached files.

Reviewer #1: No

Reviewer #2: No

Reviewer #3: No

---

## [Author Response · Author response to Decision Letter 0]

20 Apr 2020

Editor:

Answer: Thank you for your kind reminding. We have carefully modified the style of the manuscript by referring to the provided template.

Answer: Thank you for your kind reminding. We have updated the Data Availability Statement as required. All the original data used in this study was downloaded from the public database and the code used to reproduce the results was provided as supplementary files.

Answer: We have revised the cover letter as required. There is no ethical or legal restriction on sharing the datasets used in this study. The data sources and detailed methods used to reproduce the results of this study have been indicated in the manuscript and supplementary files.

3. Please include a copy of Table 1 and 2 which you refer to in your text on page 7.

Answer: We have added the Tables to the end of the manuscript as required.

Answer: We have added the captions of supplementary files in the manuscript.

Reviewer #1: In this paper by Jin et al, the authors evaluated circulating microRNA signatures from patients with soft tissue sarcomas (STS) in order to develop predictive models of diagnosis. The investigators evaluated more than 880 sarcoma samples and determined 13 gene pairs that could be used for diagnosis with an AUC of 100% in their initial set, but only 68% in the external validation set. Of note, they also determined that the miRNA gene pairs showed enrichment for functional roles in cell cycle regulation. They concluded that their serum miRNA pairs can be used for the diagnosis of STS with high degree of accuracy and sensitivity. Overall, these authors have developed a non-invasive and potentially clinically applicable prediction model using serum microRNA signatures. However, there are some concerns regarding their approaches and conclusions, which are mentioned below.

Lines 59-60: Please further explain exactly what is meant by the statement: “detecting the level of miRNAs in body fluids, the state of somatic tumor cells can be inferred..”

Answer: We really appreciate for your hard work. We further explained it in the manuscript. (lines 54 - 62). 

Lines 106-107: What does it mean for the miRNAs to be “paired in the order from front to back”?

It is still not clear how the miRNAs were paired. It would be helpful to more of the general audience to further explain this process.

Answer: Thank you for your suggestions. We first arranged the genes in ascending order according to their names in the expression matrix, and then made non-repeating pairings. Since the orders within gene pairs has no effect on the prediction model, we removed this section of description. Instead, we provide the code for building gene pairs in the supplemental files. (S1 File)

Lines 159-160: How were the 96 gene pairs, from more than 11,0000 pairs, selected to construct the prediction model.

But then in lines 163-164 it is stated that a total of 13 gene pairs, containing 16 miRNAs were used. All of this process is very confusing and difficult to follow for the reader.

Answer: We apologize for the inconvenience caused by our unclear description. We optimized the process and used the t test to screen out genes and gene pairs that had significant differences between healthy and malignant sarcoma samples. (lines 93 - 109)

Lines 173-174: It would be informative to provide more information regarding the other models used for prediction development and the comparison with this particular model.

Answer: Thank you for your kind suggestions. We compared the gene pair classifier with the gene expression based LDA model of Asano. (line 187-188, 232-234)

Finally, all of the figures are extremely difficult to review. The clarity is extremely poor.

Answer: We apologize for that mistake. We have reorganized all the figures to solve this problem.

Reviewer #2: In this manuscript, the authors describe a miRNA gene pairs-based index to differentiate patients who have bony lesions from healthy controls. The concept is interesting and such biomarker development has tremendous promise in the sarcoma field. In particular, it is intriguing that the predictive signature was consistent across all stages of disease. However, there are critical weaknesses that need to be addressed for this manuscript to be considered for publication.

Major critiques:

1. The patient cohort is comprised of tumors listed as benign, intermediate, and malignant, all of which (individually and collectively) can be distinguished from healthy controls. Benign bone lesions (40+% in the cohort), by definition, are not sarcomas, and therefore the major conclusion that this model can distinguish sarcoma is not substantiated by the data. The signature can impressively distinguish all bone lesions from healthy controls in the test/training set, and there is only a minimal ability (figure 3C) to distinguish malignant vs benign. The authors may want to contend that a benign bone lesion may undergo malignant transformation, and therefore distinguishing benign vs intermediate vs malignant is not critical in their model.

Answer: We really appreciate for your genius ideas. According to your idea, we tried to build a model to distinguish benign and malignant sarcomas, but the result was poor. These results were consistent with the conclusions of Asano's paper, and the two types of tumors could not be distinguished by PCA of miRNAs. But with reference to your ideas, we reselected the samples. Only healthy samples and malignant samples were retained, and the model was built. Our model showed high specificity in both external and internal data sets and has potential for clinical screening and complementary diagnosis.

2. The title, abstract, and body of the manuscript state that the approach has been validated, yet the validation in a very small (25 patient) external dataset did not perform nearly as well as the test/training set (65%). This is not unusual performance for such assays when there are differences in sample processing, chips, etc, though it fails to achieve the bar of validation as highlighted in the title and elsewhere. Of interest, the external dataset contained patients with varied sarcomas (i.e., malignant tumors), yet the test/training sets were composed of patients who had malignant and benign lesions. Separating out benign from malignant, and possibly eliminating patients with “intermediate” tumors from your analysis may lead to a more robust predictive model.

Answer: Thank you for your kind suggestions. By eliminating both intermediate and benign tumors from the dataset, we have constructed a more robust predictive model for sarcoma discrimination.

3. The broad title focused on miRNA signatures for sarcoma does not reflect the findings in the manuscript since it’s not only sarcoma that was differentiated from healthy controls, but rather bone lesions that may or may not be sarcoma.

Answer: Thank you for your reasonable concern. In order to make the experiment design more scientific, benign and intermediate samples were eliminated in this version of manuscript.

Minor critique:

1. In the introductory background about the promise of early detection of patients with sarcoma from healthy controls, it would be more compelling to describe practical application, such as a patient presenting with a bone lesion and this model signature can differentiate benign from malignant. Further clinical application of such an assay could be as a post-treatment screening test for patients with sarcoma to identify early recurrence of disease. However, based on the major critiques above, this may not be a potential application of the described research.

Answer: It was a good idea, and we tried it. In the data set of Asano, only samples of benign and malignant were retained, and these samples were randomly divided into training sets and verification sets in a ratio of 3:1. Using gene pairs to build models in the training set can achieve more than 90% accuracy in the validation set. However, due to the lack of external data sets, we could not find other sequencing data of serum miRNA samples from benign and malignant samples in the database, so we could not evaluate the performance of the model in external data. In the present version of manuscript, we modified the datasets and methods for model construction. The model was able to distinguish sarcoma samples from healthy samples in external data sets with an accuracy of 74.3%.

Reviewer #3: This manuscript utilizes an in silico analysis of miRNA database to develop a predictive model for the detection of sarcomas from serum samples.

Some suggestions for the authors consideration:

1. Line 160, the authors states that 96 gene pairs were selected to construct the prediction model, however in the prediction model the authors state that a total of 13 gene pairs were used. The authors should consider in explaining further as to how they further narrowed down the gene pairs from 96 to 13 and perhaps also why they felt that was necessary.

Answer: Thank you for your reasonable concern. In the actual modeling process, we tested a series of parameters, aiming to reduce the number of genes needed as much as possible while ensuring the accuracy of the model. In the current version of manuscript, we've optimized this process to make it easier to understand.

2. In Line 178, the authors state that the gene pair prediction model showed an AUC of 68.4% in the outside data set. This is a large drop in the ability of the model to predict sarcoma when using an external data set. In some literature an AUC of 0.7-0.8 is considered acceptable and 0.8-0.9 is considered excellent. The authors conclusion that their model has high accuracy and sensitivity does not seem to be consistent when they tested their model to the external data set. The AUC of 100 in the internal data set, may simply denote that differences in the methodologies used to measure the serum miRNA may have a larger impact than the gene-pair model they developed. The authors should explain the implications of the drop in the AUC in regards to the broader applicability of the model.

Answer: Thank you. In order to reduce this gap and improve the accuracy of the model in the external data set, we restrict the data set, only healthy and malignant samples were retained. In the present version of manuscript, the model achieved an accuracy of 74.3% in the external dataset with a specificity of 90%.

3. The authors conclude that their test is very sensitive. It would be very important for the authors to better state the stage of the disease when the serum samples were obtained. If the serum samples are all obtained when the patients have active detectable disease, what would be the applicability of this model? Knowing that a patient had a "sarcoma" may not necessarily be practically beneficial, as a biopsy would still be required to define histological type. In lines 239-240, the authors state that this model "is promising in the diagnosis of sarcoma". However, that information, even if the AUC was better than 68% would not be sufficient for making treatment decisions. How well does this model work in the minimal residual disease state? This may have broader applicability to replace serial imaging? The authors may consider narrowing their conclusions to state that the model they have generated was outstanding in their internal test data set, but had significant reduction in the AUC when exposed to an internal data set in detecting sarcomas in patients with known active disease.

Answer: Thank you for your kind suggestions. In the present version of manuscript, we provided the stage information in Table 1. As you can see, early stage patients (stage Ⅰ and stage Ⅱ) accounts for 40% in the dataset. It indicated that our gene pair based classifier could identify sarcomas of all stages. We narrowed our conclusions as suggested (line 1,29,49,248). We would like our gene pair based classifier to be a screening method for sarcoma.

4. in the methods section for the sake of completeness, the authors may consider adding on which platform the statistical analysis was performed.

Answer: Thank you for your kind suggestions, we have described the software and statistical methods used in our analysis as recommended. (line 143-152)

---

## [Decision Letter · Decision Letter 1]

3 Jun 2020

PONE-D-19-34131R1

A novel serum miRNA-pair classifier for diagnosis of sarcoma

PLOS ONE

Dear Dr. Zhu,

Thank you for submitting your manuscript to PLOS ONE. After careful consideration, we feel that it has merit but does not fully meet PLOS ONE’s publication criteria as it currently stands. Therefore, we invite you to submit a revised version of the manuscript that addresses the points raised during the review process.

Please address the 2 minor concerns raised by Reviewer 2.

We look forward to receiving your revised manuscript.

Kind regards,

David M Loeb

Academic Editor

PLOS ONE

Reviewers' comments:

Reviewer's Responses to Questions

**Comments to the Author**

1. If the authors have adequately addressed your comments raised in a previous round of review and you feel that this manuscript is now acceptable for publication, you may indicate that here to bypass the “Comments to the Author” section, enter your conflict of interest statement in the “Confidential to Editor” section, and submit your "Accept" recommendation.

Reviewer #2: All comments have been addressed

Reviewer #3: All comments have been addressed

2. Is the manuscript technically sound, and do the data support the conclusions?

Reviewer #2: Yes

Reviewer #3: Partly

3. Has the statistical analysis been performed appropriately and rigorously? 

Reviewer #2: Yes

Reviewer #3: I Don't Know

4. Have the authors made all data underlying the findings in their manuscript fully available?

Reviewer #2: Yes

Reviewer #3: Yes

5. Is the manuscript presented in an intelligible fashion and written in standard English?

Reviewer #2: Yes

Reviewer #3: Yes

6. Review Comments to the Author

Reviewer #2: The authors have addressed reviewer critiques. An additional limitation of the study that was not addressed in the discussion is pooling of all stages of sarcoma, as this may skew results/accuracy/specificity in that higher stage patients may have more biologically aggressive disease and circulating tumor biomarkers. Subset analyses, if numbers were high enough, would be an additional important part of future investigation and validation of the signature.

Reviewer #3: the authors have addressed all the reviewer's comments. However, they have changed some language, which may require further clarification.

1. Statistical analysis: in the original manuscript, the authors used AUC. In this version, the authors changed the analysis to accuracy. They should explain in the statistical analysis how they define accuracy. They also should include that they are going to describe sensitivity, specificity, negative and positive predictive values in the statistical section.

2. Results: characteristic datasets. In the original manuscript the authors had included intermediate and low grade tumors. As per the authors, they have excluded these from this analysis. They may want to clarify in the characteristics of datasets section that they only included high-grade lesions.

7. PLOS authors have the option to publish the peer review history of their article (what does this mean?). If published, this will include your full peer review and any attached files.

Reviewer #2: No

Reviewer #3: No

---

## [Author Response · Author response to Decision Letter 1]

7 Jun 2020

1. Statistical analysis: in the original manuscript, the authors used AUC. In this version, the authors changed the analysis to accuracy. They should explain in the statistical analysis how they define accuracy. They also should include that they are going to describe sensitivity, specificity, negative and positive predictive values in the statistical section.

Answer: Thank you for your kind suggestions. In this study, accuracy=(TP+TN)/(TP+TN+FP+FN)

(TP: True positive; TN: True negative; FP: False positive; FN: False negative), we have explained it in statistical analysis as suggested (line: 152-156) and expanded the results (line:187-193).

2. Results: characteristic datasets. In the original manuscript the authors had included intermediate and low grade tumors. As per the authors, they have excluded these from this analysis. They may want to clarify in the characteristics of datasets section that they only included high-grade lesions.

Answer: Thank you for your kind reminding. We have clarified it in the characteristics of datasets section as suggested (line: 160).

---

## [Editor Report · Decision Letter 2]

30 Jun 2020

A novel serum miRNA-pair classifier for diagnosis of sarcoma

PONE-D-19-34131R2

Dear Dr. Zhu,

We’re pleased to inform you that your manuscript has been judged scientifically suitable for publication and will be formally accepted for publication once it meets all outstanding technical requirements.

Kind regards,

David M Loeb

Academic Editor

PLOS ONE
---

## [Editor Report · Acceptance letter]

6 Jul 2020

PONE-D-19-34131R2 

A novel serum miRNA-pair classifier for diagnosis of sarcoma 

Dear Dr. Zhu:

I'm pleased to inform you that your manuscript has been deemed suitable for publication in PLOS ONE. Congratulations! Your manuscript is now with our production department. 

Kind regards, 

on behalf of

Dr. David M Loeb 

Academic Editor

PLOS ONE